# Melt Spinning of Flexible and Conductive Immiscible Thermoplastic/Elastomer Monofilament for Water Detection

**DOI:** 10.3390/nano12010092

**Published:** 2021-12-29

**Authors:** Julie Regnier, Aurélie Cayla, Christine Campagne, Éric Devaux

**Affiliations:** GEMTEX–Laboratoire de Génie et Matériaux Textiles, ENSAIT, Univ. Lille, F-59000 Lille, France; julie.regnier@ensait.fr (J.R.); christine.campagne@ensait.fr (C.C.); eric.devaux@ensait.fr (É.D.)

**Keywords:** water leak detection, conductive polymer composite, immiscible thermoplastic/elastomer blend, carbon nanotubes, filament

## Abstract

In many textile fields, such as industrial structures or clothes, one way to detect a specific liquid leak is the electrical conductivity variation of a yarn. This yarn can be developed using melt spun of Conductive Polymer Composites (CPCs), which blend insulating polymer and electrically conductive fillers. This study examines the influence of the proportions of an immiscible thermoplastic/elastomer blend for its implementation and its water detection. The thermoplastic polymer used for the detection property is the polyamide 6.6 (PA6.6) filled with enough carbon nanotubes (CNT) to exceed the percolation threshold. However, the addition of fillers decreases the polymer fluidity, resulting in the difficulty to implement the CPC. Using an immiscible polymers blend with an elastomer, which is a propylene-based elastomer (PBE) permits to increase this fluidity and to create a flexible conductive monofilament. After characterizations (morphology, rheological and mechanical) of this blend (PA6.6_CNT_/PBE) in different proportions, two principles of water detection are established and carried out with the monofilaments: the principle of absorption and the short circuit. It is found that the morphology of the immiscible polymer blend had a significant role in the water detection.

## 1. Introduction

In the last decade, the development of new detectors in the smart textile fields has been increasing. They are used to control physical parameters like for instance temperature variation [1], stress and deformation [2] for athletics or health fields or presence of liquids and gases [3,4,5,6,7] in industry and environmental protection [8]. This technology of smart textiles is based on the electrical conductivity variation of the material in interaction with its environment to detect and transmit the data. In the field of monitoring, more and more construction industries use composites to reinforce their concrete structures as for instance retention tanks. Concrete is a fragile material, which can cause fluid leakages, and which is sometimes dangerous for the environment (pollutant). Smart textiles can be incorporated inside the composite resin or concrete to monitor the stress and deformation of the structure and detect a fluid leakage in case of damage. Consequently, more and more researches develop intelligent composite membranes [9,10,11,12]. The intelligent composite membranes are composed of two linked parts: the matrix, which is the resin, or the concrete reinforced by textile structure including smart filaments, which detect problems.

To detect fluids with smart textiles, different technologies that are based on the electrical conductivity variation of the textile in contact with the fluid can be employed. The most commonly studied in the literature are the intrinsically conducting polymers (ICPs) [13,14,15] or the conductive polymer composites (CPCs) [16,17,18]. They have the particularity of modifying their electrical conductivity according to the affinity of the polymer with the fluid [18,19,20]. They can be presented in the form of films [21], monofilaments [22], multifilaments [6,20], or coating [23]. Regarding the ICPs, the detection of fluid results in the electrical conductivity variation of the IPC due to oxidation-reduction or acid-base reactions in the presence of the fluid [24]. Dzugan et al. [25] have worked on semiconductive organic materials. They have observed the electrical conductivity variation of a phthalocyanine thin layer on an electrode in contact with different vapors. They have revealed the good sensitivity to the humidity contrary to other analyte vapors (ethanol, acetone, toluene, chloroform). Regarding the CPCs, they are formed from a matrix of polymer in which a sufficient quantity of fillers is incorporated to develop the yarn conductive. This minimum charges concentration, which is introduced to provide the electrical property, is called the percolation threshold [26]. A strong affinity between the polymer and the fluid allows this one to diffuse inside the CPC. This absorption phenomenon causes the swelling of the matrix and, consequently, the variation of the interparticular distances, which induces the CPC electrical conductivity variation. This property of absorption to detect fluids is not yet commonly used in industry but is more at the research stage in the relevant literature. In their study, Castro et al. [4] have analyzed different polymers (poly (methyl methacrylate) (PMMA), polycarbonate (PC), polycaprolactone (PCL), polylactic acid (PLA), and biosourced poly (ester) (BPR)) filled with carbon nanotubes (CNT) in a form of film, which is deposited with a layer-by-layer spray on interdigitated electrodes. The electrical signals were investigated for different vapors like water, ethanol, dichloromethane, and toluene. They have noticed that the polymers have different sensitivities according to the chemical nature of solvent vapors. For instance, the PCL has a better sensitivity with the tetrahydrofuran vapor while the PC detects better the dichloromethane than the other vapors. They have also observed that the humidity is better detected by the PCL or the PMMA. More and more researchers blend different polymers in the CPC to combine their properties [27] or improve the compounds preparation [1,28]. For instance, in the study of Segal et al. [29], the filled blend of a thermoplastic polymer and a thermoplastic elastomer permits the improvement of the fillers networks and, thus, of the detection of different alcohols. The morphology of an immiscible polymers blend depends on different parameters: the proportion of the blend’s components [30], the extrusion parameters such as the shear conditions [31], the viscosity and the elasticity of the blend [32,33]. It can have a disperse phase in another or a co-continuity of the phases. The continuity of the fillers in one phase and the continuity of each phase can maintain the electrical property while reducing the fillers content. Therefore, this morphology phenomenon is the double percolation [34].

The other principle of detection is the short circuit method. It is based on two conductive yarns in parallel without initial contact. For the detection, the liquid has to create a conductive connexion between the two conductive yarns [35,36]. The signal depends on the properties of the yarns and the liquid but also of the distance between two conductive yarns and the structure of the textile [37].

In this study, an immiscible polymers blend of a filled thermoplastic polymer and an elastomer is developed, characterized, and investigated for the water detection. The polyamide 6.6 (PA6.6) has been already investigated for a humidity sensor [38] with glass fiber in the reinforced textile for the composite [39,40,41]. With its good affinity with water [42], this polymer is chosen for the water detection property. Thanks to the previous study of Javadi Toghchi et al. [43], 2 wt.% of multiwall carbon nanotubes (MWCNT) incorporated in the PA6.6 thermoplastic polymer is needed to exceed the percolation threshold. A polypropylene-based elastomer (PBE) is added to the blend by a second extrusion in different proportions to add enough flexibility [27] to not break inside the resin when it cracks. Therefore, it permits to improve the mechanical property of the water detector filament [44]. The elastomer permits also to increase the blend fluidity (rheological property) to improve the implementation by melt spun [45]. The study of the monofilament morphology influence on the rheological, mechanical, and water detection properties permits to identify the co-continuous blend. Two principles are investigated for the water detection: the principle of absorption and the short circuit.

## 2. Materials and Methods

### 2.1. Materials and Process

#### 2.1.1. Conductive Yarns

The monofilaments are composed of a blend of a thermoplastic polymer with fillers and a propylene-based elastomer (PBE).

The first blend is the thermoplastic polymer with 3 wt.% of fillers. The thermoplastic polymer is the polyamide 6.6 (PA6.6) TORZEN U4803 NC01 produced by Invista (Wichita, KS, USA), which has a melting point of 263 °C. The fillers are the multiwalled carbon nanotubes NC 7000 (MWCNT) supplied by Nanocyl (Sambreville, Belgium). These MWCNTs have an average length of approximately 1.5 μm, a diameter of 9.5 nm, and a specific area of 250–300 m^2^/g. The polyamide 6.6 has a moisture regain of about 4% [46]. Therefore, thanks to the PA6.6’s affinity to water, this blend is used for the detection by the electrical conductivity variation.

The second blend is the addition of the PBE to the PA6.6_3CNT_ to increase the fluidity of the blend and, thus, to facilitate the compounds preparation [47]. The PBE is also used to add a flexible property to the detection yarn to not break during the resin cracking. The employed elastomer is the VISTAMAXX 3000, which is supplied by ExxonMobil Chemical (Houston, TX, USA).

#### 2.1.2. Compounds Preparations

Two successive extrusions are realized in order to obtain detector yarns. First, the incorporation and the dispersion of 3 wt.% of MWCNT in the PA6.6 (PA6.6_3CNT_) are processed by a co-rotating intermeshing twin-screw extruder from Thermo-Haake PTW 16/25p (barrel length = 25:1 L/D). The second step allows to add different percentages of PBE (from 0 to 50 wt.%) in the blend (Table 1). The second extrusion use the Process 11 Parallel Twin-Screw Extruder from Thermofischer (Waltham, MA, USA) with a barrel length of 40:1 L/D.

The processing conditions was based on the study of Javadi Toghchi et al. [43], which have already worked on the extrusion of the PA6.6_3CNT_. The rotating speed of these extruders is 100 RPM, and the temperatures profiles are reported in the Table 2. Before each extrusion, the polymer pellets are dried at 80 °C for 16 h.

The monofilaments have a diameter of approximately 1.5 mm ± 0.07 mm.

### 2.2. Methods of Characterisation

#### 2.2.1. Methods of Characterization of the Morphology and the Mechanical and Rheological Properties

##### Morphology’s Characterization

For the Scanning Electron Microscope (SEM) images, the samples are prepared by cryofracture. The samples are frozen under nitrogen and then broken. They are observed by a Schottky Field Emission Scanning Electron Microscope SU5000 at 5 kV and different magnifications.

For the Transmission Electron Microscopy (TEM) images, the samples are prepared by the ultramicrotomy method to have clean and flat surfaces. The different polymer phases are detectable thanks to the CNT present in the PA6.6.

##### Rheological Properties Characterization

The Melt Flow Index (MFI) determines the flow ability of a polymer and more precisely the flowing polymer weight in 10 min at a certain temperature. In this study, the test is executed on the melt flow tester from Thermo-Haake with a temperature of 270 °C and a pressure of 2.16 kg according to the standard ISO-11333. Before the test, the polymer pellets are dried at 80 °C for 16 h.

##### Mechanical Property Characterization

The elongation at break of monofilaments is measured by an MTS Criterion tensile bench from MTS (Minnesota, USA). The tests are realized with an initial length of 150 mm, initial speed of 500 mm/min and a pre-loaded of 5 N. They are executed under a controlled and conditioned atmosphere of 65% of relative humidity and a temperature of 20 °C. Five measurements for each blend are necessary to accept the results with a standard deviation of about 20%.

#### 2.2.2. Water Detection Methods

To validate the results and the repeatability of the protocols, all the water detection tests are realized ten times for each monofilament and conditioned at a room temperature of 20 °C and a relative humidity of 35%. Since the detection’s mechanisms depends directly on the ionic conductivity of the demineralized water, this parameter is controlled before each test with a conductimeter (Tacussel electronic type cdrv 62) and a conductivity standard solution of 1413 µS/cm at a temperature of 25 °C (KCL, Fischer Scientific, Waltham, MA, USA). The surface tension of the water is also overseen and measured by the tensiometer “3S Scales” from GBX Instruments. The demineralized water had an ionic conductivity of about 4.7 ± 0.9 µS/cm and a surface tension of about 71.8 ± 3.2 mN/m.

For all the electrical conductivity tests, the monofilaments are connected to an a Keithley 2461 SourceMeter (Beaverton, OR, USA). This device measures the current intensity while applying a voltage which ranges from −0.5 V to 15 V with an increment of 0.1 V. Using the data of the current as a function of the voltage sent, the electrical conductivity for a length of 10 cm is determined by the Equation (1). The inverse of the resistance is the directing coefficient of the linear trendline of the current-voltage functions between 0 and 12 V.
(1)$$\sigma \;\left(S/m\right)=\frac{L}{RS}$$
where *σ* is the electrical conductivity of the system, *L* is the monofilament’s length (*L* = 0.1 m), *S* is the monofilament’s area (m^2^) and *R* is the resistance measured (Ω).

The short circuit is based on a conductive path creation between two parallel monofilaments through the drop of water (Figure 1a). The electrical signal is detected when the water makes the link between the two electrically conductive filaments. To compare the different proportions of the PBE in the blend, the conductance is calculated (Equation (2)) from the measured resistance. A square of 3 × 3 cm of absorbent paper is added on the parallel monofilaments to deposit the drop of water to make the test repeatable. With absorbent paper, the spreading of the drop is controlled by eliminating the shape of the drop and the problems of absorption that could vary the conductance of the circuit:(2)$$G\;\left(S\right)=\frac{1}{R}$$
where *G* is the conductance of the circuit (*S*) and *R* is the resistance of the circuit (Ω).

The principle of detection by absorption (Figure 1b) is based on the modification of the resistance of the yarn when it is soaked in water. The electrical conductivities of the dry and wetted monofilament are calculated (Equation (1)). To observe the influence of the PBE percentage on the detection, the detector sensitivity (*Sw*) calculated corresponds to the change in the electrical conductivity between the dry and the wetted monofilament (Equation (3)):(3)$$S_{w}\;\left(\%\right)=\;\frac{\left(\sigma w-\sigma d\right)}{\sigma d}\times 100\;$$
where *Sw* is the detector sensitivity (%), σd represents the dry electrical conductivity (S/m), and *σw* is the wetted electrical conductivity (S/m).

To find the optimal formulation for the CPC detector filament, a figure of merit is defined. It considers the mechanical and detection properties (Equation (4)):(4)$$F=\frac{e\times S_{w}}{100}$$
where *F* is the figure of merit (%), e is the elongation at break (%) for the mechanical property, and *Sw* the filament sensitivity to water (%).

## 3. Results and Discussion

### 3.1. Impact of the PBE Proportions on the Morphology and the Mechanical/Physical Properties

#### 3.1.1. Morphology of the Blends

The visualization of the SEM and TEM images are important to validate the different assumptions of the blends’ behaviors. By hypothesis, the continuity of the PA6.6_3CNT_ phase is influenced by the blend proportions and the localization of the CNT in the PA6.6 is favored by the separation of the compounds process into two extrusions.

The observation of TEM image of the PA6.6_3CNT_ permits to conclude on the good dispersion of the CNT (black color on the Figure 2a) in the PA6.6 polymer (lighter background on the Figure 2a).

Regarding PA6.6_3CNT_90/PBE10 blend, the SEM image (Figure 2b) reveals a nodular morphology of the PBE in the PA6.6_3CNT_. Moreover, no migration or aggregate of the CNT in the PBE or at the interface between the two polymers are detected with the TEM image (Figure 2c). This kind of morphology and this CNT localization can be confirmed in other studies [46,48,49].

Regarding the PA6.6_3CNT_60/PBE40 blend, the SEM image (Figure 2d) indicates a fibrillar morphology of the PA6.6_3CNT_ in the PBE. Therefore, the addition of 40 wt.% of PBE in the blend permits to achieve a phase inversion (Figure 3). The majority phase of PBE becomes the CPC matrix, which modify the monofilaments properties. Moreover, thanks to the TEM image (Figure 2e), the CNT (black) have not migrated into the PBE (lighter color) or aggregated at the interface as the PA6.6_3CNT_90/PBE10 blend.

#### 3.1.2. Rheological Properties

The influence of the proportion of CNT and PBE on the rheological properties is investigated through the Melt Flow Index (MFI) analysis (Figure 4). The CNT decreases the blend fluidity whereas the PBE permits to increase it.

The addition of PBE does not improve the fluidity of the blend without fillers: 73 g/10 min for the PA6.6 and 66 g/10 min for PA6.6 blended with 50 wt.% of PBE. However, it permits to overcome the decrease of the fluidity due to the CNT: from 9.5 g/10 min for the PA6.6_3CNT_ to 26 g/10 min for the PA6.6_3CNT_50/PBE50, so the fluidity of the filled blend increases with the proportion of elastomer. Therefore, it is found that the CNT increases the viscosity of the blend contrary to the PBE, which increases the fluidity of the polymers blend. The addition of CNT creates more links between the fillers and, thus, reduces the mobility of the macromolecular chains of the PA6.6 [43,48]. The PBE is, thus, adding to overcome to this high viscosity and to improve the blend implementation by melt spun [27].

#### 3.1.3. Mechanical Property

The elongation at break is also studied to observe the influence of the fillers and the elastomer on the blend (Figure 5). The addition of CNT in the blend decreases weakly the mechanical property [49] contrary to the PBE addition. The PBE permits to increase the elongation at break of the filled blends until the phase inversion.

Regarding the unfilled blends, the addition of 10 wt.% of PBE permits to increase of 23% the monofilament elongation. Beyond 10 wt.%, the weak interfacial cohesion between PA6.6 and PBE, which increases with the PBE proportion, leads to a decrease in elongation.

Regarding the filled blend, the elongation at break of the monofilaments with less than 30 wt.% of PBE slightly varied from 16 to 20%. While, after 30 wt.% of PBE, the mechanical property abruptly decreases: to 4%. This mechanical property variation can be correlated with the morphology variation. Before 30 wt.%, the nodules of PBE permits a larger elongation of the monofilament before the break. The interfacial area between the two polymers has a low cohesion. Therefore, with a high percentage of PBE, the interface between the PA6.6_3CNT_ fibrils and the PBE increases and causes the premature break of the monofilament. This result is confirmed in the study of Qiu et al. [50] on a polyamide 6/polyolefin elastomer blend. The tensile strength property decreases with the addition of elastomer in the blend. They have explained that this result was expected due to the morphology of the blend and the lower tensile strength of elastomer compared with polyamide 6. Theses hypotheses are also verified with other studies on the influence of the elastomer on polycarbonate/CNT blend [45,51].

### 3.2. Electrical Propreties and Water Detection

#### 3.2.1. Electrical Properties

The initial electrical conductivity depends on the morphology of the blend: the localization and the dispersion of the CNT and the morphology of the polymer phases.

The electrical conductivity decreases with the adding of PBE in the blend (Figure 6): from 1.2 × 10^−2^ S/m to about 2.6 × 10^−3^ S/m. With a nodular morphology, the dry conductivities are approximately the same: from, respectively, 2.6 × 10^−3^ S/m to 1.5 × 10^−3^ S/m for 10 wt.% and 30 wt.% of PBE. However, the dry conductivity decreases suddenly to 2 × 10^−7^ S/m with the percentage of PBE above 30 wt.%. The phase inversion between 30 and 40 wt.% causes the increase in the interparticular distance and, thus, the decrease of the conductivity.

#### 3.2.2. Principle of Short Circuit

The short circuit’s signal depends on several parameters: the distance between the two parallel monofilaments and the water properties, which are both fixed, as well as on the monofilaments’ properties. In this study, the conductance of the short circuit depends on the dry conductivity of the monofilament and, thus, the proportion of PBE in the blend (Figure 7). The detection signal is better when the dry conductivity of the monofilament is high and, therefore, when the proportion of PBE is small in the blend. The signal is from about 1.4 × 10^−7^ S without PBE to 8.3 × 10^−9^ S with 30 wt.% of PBE in the blend, and it decreases to 1.7 × 10^−12^ S with 50 wt.% of PBE in the blend.

#### 3.2.3. Principle of Absorption

The water detector sensitivity (Sw) of the principle of absorption is based on the variation of the yarn’s conductivity. It is the change between the electrical conductivity of the dry monofilament (dry conductivity) and of the wetted monofilament (wetted conductivity). The sensitivity to water depends on the blends’ formulation. Therefore, it has the same trend as the initial electrical conductivity (Figure 8). The blends with a high dry conductivity have a positive sensitivity: from 43 ± 13 to 28 ± 20% with the percentage from 0 wt.% to 30 wt.% of PBE. However, the blends with a low one have a negative sensitivity: respectively, −26 ± 22 and −51 ± 30 % for 40 and 50 wt.% of PBE. 

By comparing the *Sw* of the different blends, it is possible to make hypotheses regarding the percentage of PBE on the *Sw* (Figure 9). Regarding the positive *Sw*, the absorbed water increases the monofilament electrical conductivity by increasing the number of conductive paths between the CNT (Figure 9a). The negative *Sw* is due to the blend morphology inversion and with the presence of water. By hypothesis, the water permits the swelling of the PA6.6_3CNT_ fibrils, resulting in the increase of the distance between the CNT, which reduces the conductivity of the monofilament (Figure 9b).

To develop the filament detector, a trade-off between good rheological and mechanical properties and a good sensitivity is needed. The figure of merit (F), which quantifies the mechanical and detection property, decreases slightly before dropping sharply for the blends with more than 30 wt.% PBE (Figure 10). As all the properties, this fall corresponds to the phase inversion of the morphology between nodular and fibrillar. The figure of merit highlights the decrease of the elongation at break and the sensitivity to water with the addition of PBE in the blend. Moreover, to develop the detector filament using the melt spinning, the melt flow index (MFI) has to be around 20 to 25 g/10 min. The two formulations that correspond to these criteria are PA6.6_3CNT_80/PBE20 and the PA6.6_3CNT_70/PBE30. Therefore, the optimum CPC filament to optimize is revealed thanks to the figure of merit which is the PA6.6_3CNT_70/PBE30.

## 4. Conclusions

This study is focused on the development and the characterization of an immiscible thermoplastic/elastomer blend for the water detection. The monofilament is created by two successive extrusions: by filling the PA6.6 with CNT first, and then by adding the PBE in different blend proportions.

Regarding the rheological properties, the CNT increase the blend viscosity whereas the fluidity increases with the proportion of PBE. The observations have correlated the mechanical and water detection properties with the blends’ morphology. Two morphologies are revealed by the SEM and TEM images: the nodular and fibrillar morphology.

Below 30 wt.% of PBE in the blend, the PBE is in the form of nodules dispersed in the PA6.6_3CNT_. The elongation at break increases with the proportion of PBE, while the electrical conductivity slightly decreases. The sensitivity of the absorption principle and the conductance of the short circuit follow the same trend as the dry conductivity of the monofilament. The sensitivity has a positive change, which means that the conductivity of the detector monofilament increases with the water contact. It creates new conductive paths between fillers and water.

Above 30 wt.% of PBE, the morphology changes for a fibrillar form of the PA6.6_3CNT_ in the PBE. This loss of phase percolation corresponds to the decrease of the mechanical, electrically conductive and detection properties. Regarding the mechanical property, the bad interface between PA6.6_3CNT_ and PBE increases with the PBE proportion, which causes a premature break of the monofilament. Regarding the absorption principle, the sensitivity becomes negative. The electrical conductivity of the monofilament decreases with the contact with water due to the modification of the interparticular distance. To find a trade-off between good rheological property and good mechanical and water detection properties, the figure of merit shows that the best candidate to optimize is the monofilament of PA6.6_3CNT_70/PBE30. The objective is to increase and refine the sensitivity of the detection filament by passing this formulation in melt spinning. Future work aims to develop this multifilament to reduce the standard deviation of its water sensitivity and to verify its morphology and its electrical property.

## Figures and Tables

**Figure 1 nanomaterials-12-00092-f001:**
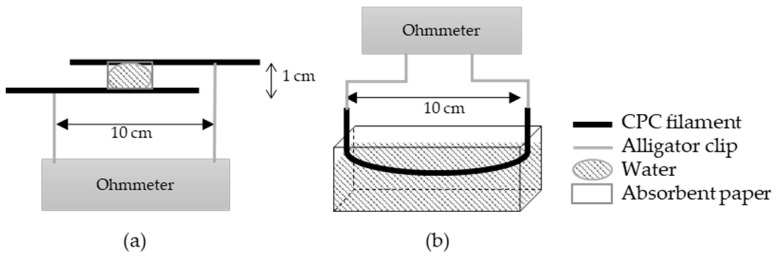
Scheme of the principle of detection by the short circuit (**a**) and by absorption (**b**).

**Figure 2 nanomaterials-12-00092-f002:**
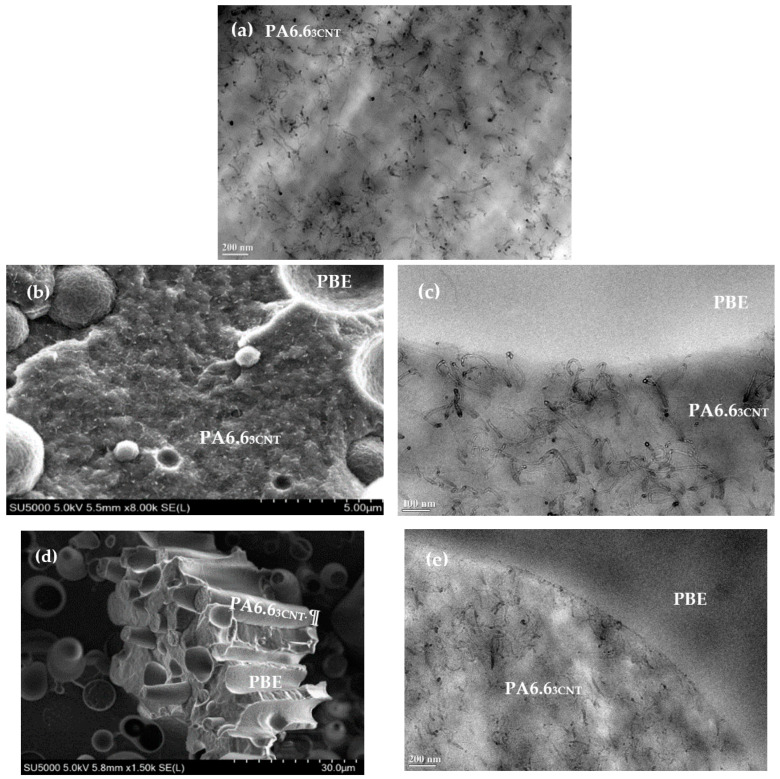
TEM images of PA6.6_3CNT_ (**a**) and SEM (left)/TEM (right) images of PA6.6_3CNT_90/PBE10 (**b**,**c**) PA6.6_3CNT_60/PBE40 (**d**,**e**) blends.

**Figure 3 nanomaterials-12-00092-f003:**
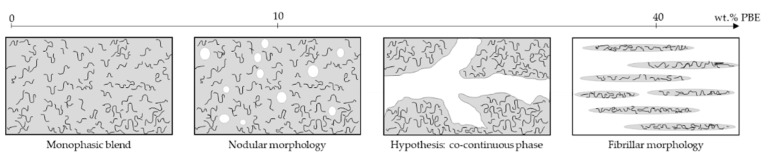
Hypothesis of the evolution of phases with the addition of PBE in the blend.

**Figure 4 nanomaterials-12-00092-f004:**
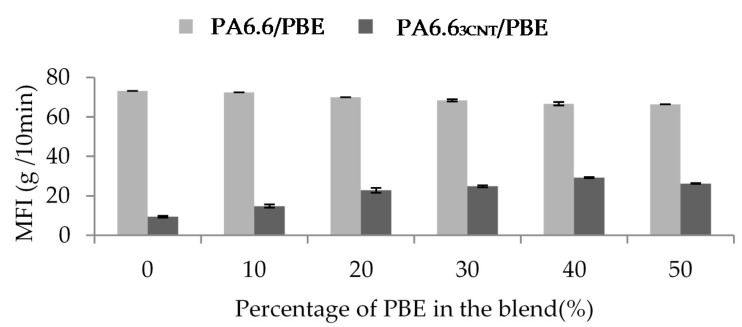
Influence of the percentage of PBE and the presence of CNT on the PA6.6/PBE blends viscosity at 270 °C, 2.16 kg.

**Figure 5 nanomaterials-12-00092-f005:**
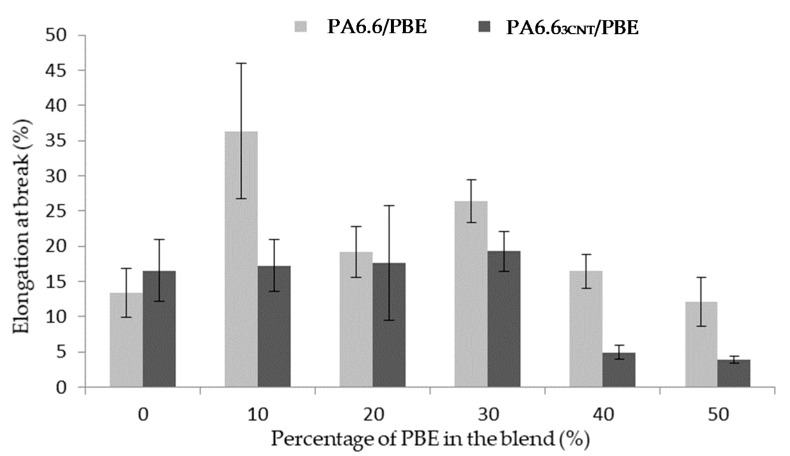
Influence of the proportion by weight of PBE on the mechanical properties of the PA6.6_3CNT_/PBE blends.

**Figure 6 nanomaterials-12-00092-f006:**
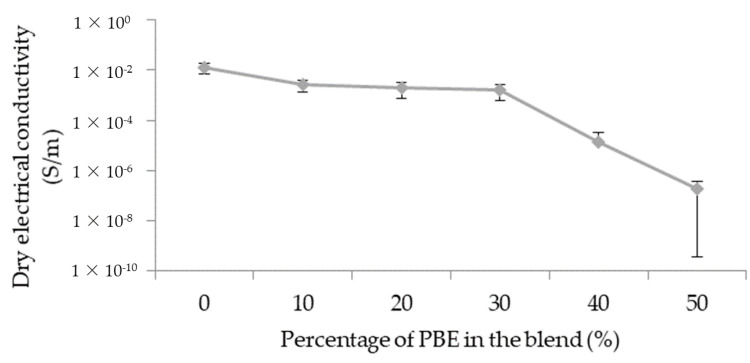
Influence of the proportion by weight of PBE on the electrical property of the PA6.6_3CNT_/PBE blends.

**Figure 7 nanomaterials-12-00092-f007:**
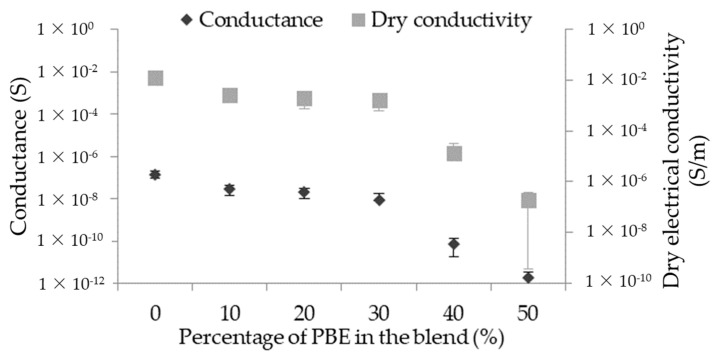
Influence of the proportion by weight of PBE on the short circuit signal of the PA6.6_3CNT_/PBE blends.

**Figure 8 nanomaterials-12-00092-f008:**
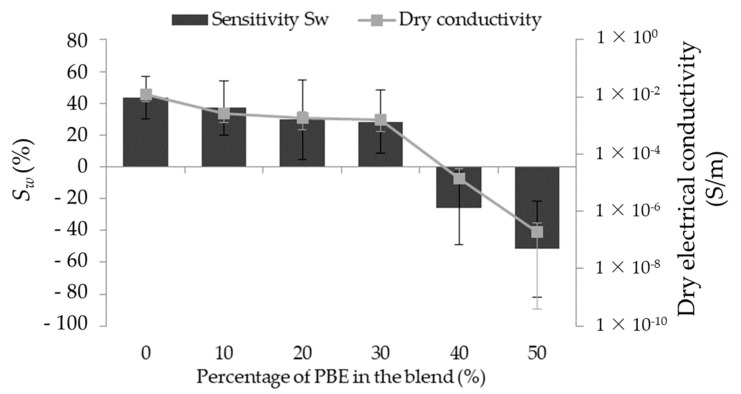
Influence of the proportion by weight of PBE on the water sensitivity of the PA6.6_3CNT_/PBE blends.

**Figure 9 nanomaterials-12-00092-f009:**
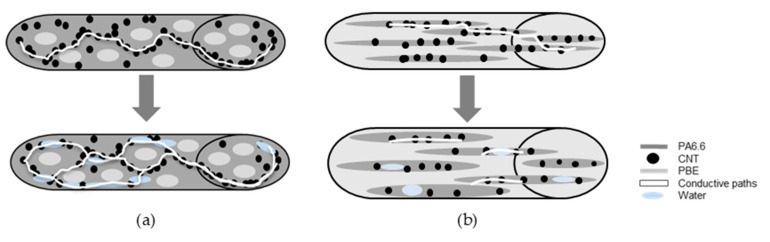
Hypotheses of the water detector monofilament: with a positive sensitivity (**a**), with a negative sensitivity (**b**).

**Figure 10 nanomaterials-12-00092-f010:**
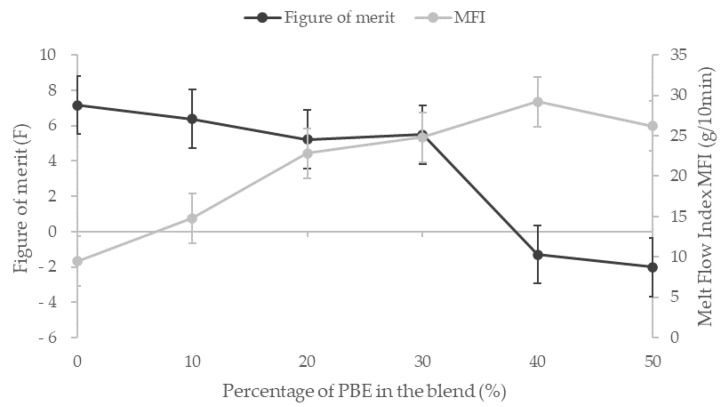
Influence of the proportion by weight of PBE on the Melt Flow Index and on the figure of merit, which gathers the elongation at break and the sensitivity of the blends.

**Table 1 nanomaterials-12-00092-t001:** Summary of the different extruded monofilaments.

Sample Reference	Blend Proportion		Total Fillers Content in the Blend
	wt.% PA6.6_3CNT_	wt.% PBE	wt.% CNT
PA6.6_3CNT_	100	0	3
PA6.6_3CNT_90/PBE10	90	10	2.7
PA6.6_3CNT_80/PBE20	80	20	2.4
PA6.6_3CNT_70/PBE30	70	30	2.1
PA6.6_3CNT_60/PBE40	60	40	1.8
PA6.6_3CNT_50/PBE50	50	50	1.5

**Table 2 nanomaterials-12-00092-t002:** Extrusion temperatures profiles.

	T1	T2	T3	T4	T5	T6	T7	T8
PA6.6_3CNT_	260	270	275	275	280	-	-	-
PA6.6_3CNT_/PBE	215	275	285	285	278	275	270	270

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
