# Peer review of "Melt Spinning of Flexible and Conductive Immiscible Thermoplastic/Elastomer Monofilament for Water Detection"

_nanomaterials, 2021, doi:10.3390/nano12010092_

Round 1

Reviewer 1 Report

Authors prepared by extrusion blends of polyamide 6.6 (PA6.6) and  propylene-based elastomer (PBE), contaning carbon nanotubes (CNT), and investigated water detection by the principle of absorption and the short circuit.

The concept behind this work is new and interesting, however, there are some points that need to be adressed in more depth:

- characterization of the polymers used, especially PBE, and the blends – please expand this section [useful references: https://doi.org/10.3390/polym11121976, https://doi.org/10.1080/00222348.2013.857536   and references therein};

- how the processing conditions were chosen / optimized?

- „Rheological properties characterization” section is limited only to MFI determination;  rotational rheometer should be applied to study the complex viscosity vs frequency, and  to discuss shear-thinning behaviour, as well as Cole-Cole plots. Importantly, the results obtained could  complete/verify the conclusions drawn from SEM/TEM;

- what is the repeatability and accuracy of the monofilament sensor PA6.63CNT70/PBE30?

- Plase add as a general reference: Ana Zubiarrain-Laserna and Peter Kruse 2020 J. Electrochem. Soc. 167 037539.

Reviewer 2 Report

The manuscript is well structured, understandable and reported results are
interesting and worth publishing. I’d recommend some improvements and I
point out two issues that should be considered by the editor.

Issue 1: In conclusions, lines 356 – 358, the authors claim that the optimum
trade off between electrical and mechanical properites is at 30% wt. of PBE.
This is difficult to follow, since both the dependence of elongation at
break (epsilon) in Fig. 5 and sensitivity (Sw) in Fig. 7 on wt% of PBE is
quite slow. Perhaps the authors should define a figure of merit, like
epsilon times Sw to quantify the trade-off and to show that the optimum is
indeed around 30 wt% of PBE. The figure of merit may also include the MFI.
(Analogously to the thermoelectric figure of merit zT, see
https://en.wikipedia.org/wiki/Thermoelectric_materials)

Issue 2: Fig. 8 and hypotheses of water detection, lines 306 – 312. The
authors explain the negative sensitivity Sw (Fig. 8b) by swelling of
PA6.6_3CNT phase that leads to increase of distances between CNTs in PA6.6 and thus to increase in the percolation threshold and decrease of the
conductivity. This mechanism must then be true also for the morphology in
Fig. 8a. Thus the positive Sw must be explained by two competing effects:
decrease of conductivity by swelling of the PA6.6 phase and increase of
conductivity by formation of new conductive paths by water between CNTs (the former decreasing the percolation threshold, the latter one increasing the percolation threshold). The measurements are done with demineralised water with conductivity of ~5e-4 S/m, while the conductivity of dry monofilaments is ~1e-3 S/m. So the new conductive paths, if they were formed solely by water, would have smaller conductivity than the dry monofilament?

-----------------------------------------------------------------------

Suggestions for improvements:

The change of the sensitivity Sw from positive to negative values happens
between 30% PBE and 40% PBE, but the morphological analysis in Fig. 2 is for 10% PBE and 40% PBE. Images for 30% PBE would be more suitable.

Section 2.2.2., lines 186 – 193
The text would be better understandable if the authors use “dry” and “wet”
instead of “initnial” and “final”. See also lines 282, 284, caption of y-axis of Fig. 6, Fig. 7. The Sw is not “rate of change” but simply change.
The rate of change would be derivative of Sw. See also lines 294, 346, 354.

Section 2.2.2., lines 178 – 182
Authors do not mention if the current-voltage relationship was linear or if
there were some deviations from Ohm’s law observable.

Section 3.1.2, line 239
Abbreviation “MFI“ should be defined.

Section 3.2.1, lines 286 – 287, do the authors mean increase in the distance
between the conductive phases (conductive particles) by “increase in the
interparticular distance”. See also line 355.

-----------------------------------------------------------------------

Language:
Line 75: “in another” instead of “in an other”
Lines 267-268: “larger” instead of  “an more”
Lines 269 – 270: “interfacial area between”  instead of “interface between”
Line 342: “Below 30 wt.%” instead of “Before 30 wt.%’”
Line 349: “Above 30 wt.%” instead of “Exceeded 30 wt.%’”

Reviewer 3 Report

This manuscript describes the elaboration of conductive monofilament and their use for water detection. The good affinity of PA6.6 with water was used to modify the electrical conductivity of composite monofilament. The electrical conductivity was provided by insertion of CNTs in PA6.6 matrix. The rate of conductive fillers increases drastically the viscosity of melt polymer. A mix with an immiscible elastomer was realized to decrease the viscosity. The electrical conductivity was measured for each rate of elastomer and exhibits a limit around 50%wt. The authors have observed by SEM the phase repartition in composites and have demonstrated a fibrillary orientation on PA6.6/CNT in elastomer matrix above 40%vol. An original result was obtained on the electrical behavior under water with opposite influence on electrical conductivity for composites filled before and above 40% of elastomer. the authors propose an interesting and original explanation based on the swelling of the PA under the effect of water

This article may be published.

Author Response

This is a perfect summary of the study highlighted in this article. Thanks you for your approval and support.

Round 2

Reviewer 1 Report

The revised manuscipt can be published as it is.